# Understanding Permutation Based Model Merging with Feature Visualizations

**Congshu Zou**[1,2]   **Geraldin Nanfack**[1,2]   **Stefan Horoi**[2,3]   **Eugene Belilovsky**[1,2]
[1]Concordia University   [2]Mila – Quebec AI Institute   [3]Université de Montréal
congshu.zou@mail.concordia.ca
{geraldin.nanfack,eugene.belilovsky}@concordia.ca
stefan.horoi@umontreal.ca

## Abstract

Linear mode connectivity (LMC) has recently become a topic of great interest. It has been empirically demonstrated that popular deep learning models trained from different initializations exhibit linear model connectivity up to permutation. Based on this, several approaches for finding a permutation of the model's features or weights have been proposed leading to several popular methods for model merging. These methods enable the simple averaging of two models to create a new high-performance model. However, besides accuracy, the properties of these models and their relationships to the representations of the models they derive from are poorly understood. In this work, we study the inner working mechanisms behind LMC in model merging through the lens of classic feature visualization methods. Focusing on convolutional neural networks (CNNs) we make several observations that shed light on the underlying mechanisms of model merging by permute and average.

## 1   Introduction

Deep neural networks have achieved remarkable advancements in recent years, demonstrating their capability to perform various valuable tasks. Understanding the inner mechanisms of neural networks becomes critical, which leads to exploring the loss landscape. Frankle et al. [6] suggest linear mode connectivity (LCM) as a paradigm where two neural networks, randomly initialized and trained using stochastic gradient descent (SGD), are observed to be connected by a linear path in the parameter space with no or little loss barrier. The study of LMC can offer valuable insights into the complex structure of loss landscapes and training dynamics. It will also help various deep learning applications such as model merging, transfer learning, etc. However, naive LMC often fails due to the permutation invariance of neural networks [4]. Thus several algorithms are designed to linearly connect two trained models [1, 10] by first aligning one to another, and second by finding a permutation. This has been shown to lead to linear mode connectivity and models that can be merged (or averaged).

However, why the LMC works with permutation models and what properties this induces on the average model is challenging to understand. In this work, we propose to study this through the lens of feature visualization. By maximizing the activation of certain neurons in a network, we can visually understand what features the model is detecting. If two models can be linearly interpolated, then they should activate similar features. In addition to visually inspecting the extracted features, we use some metrics to characterize the similarity of images extracted from two models [13].

In this study, we train multiple randomly initialized models with SGD optimization, then apply the REPAIR [10] method to permute the model weights space and visualize the top 10 images that most activate each neuron. We find that models after permutation share some similar top-activated features, but not all of them. We shed light on the features that don't match and highlight that they correspond to rare features in the models being aligned against. Finally, we show that despite the limitation of

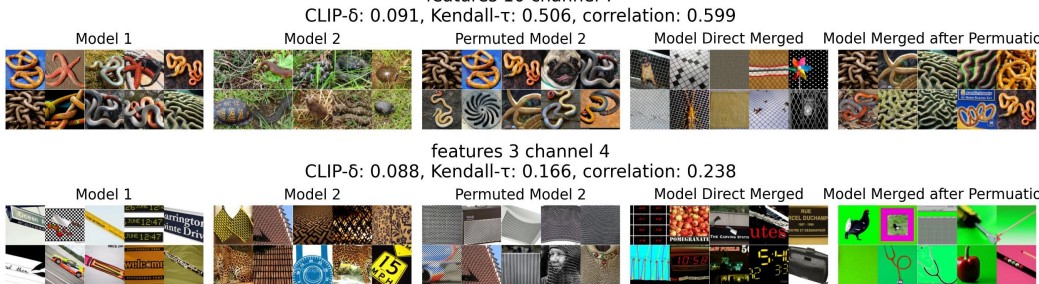

Figure 1: **Example of top images. Top** shows the permuted model 2 has similar top-activated images as model 1 and thus we have similar results in the merged model. **Bottom** shows one example that top activation images do not match in models 1 and 2. The top images from the merged model also do not match and are not from either model. More examples are included in the Appendix.

one-to-one permutation matching, features in a given model can be approximated by a sparse subset of features in another model, motivating recently proposed methods in model merging.

**Related Work**    *Linear mode connectivity*. Mode connectivity investigates the relationship between two optimal solutions found after training. It demonstrates that the optimal solutions for deep neural networks can be connected by very simple curves[7, 3]. This is further extended by Frankle et al. [6] as linear mode connectivity, which suggests that there exists a linear path between the two optima, where the loss barrier is low along the path. Later work by Entezari et al. [4] show that taking into account the permutation invariance, optimal solutions trained with SGD will likely have no barrier in the linear interpolation between them, i.e. linear mode connectivity [4]. Based on this conjecture, more recent studies have developed a few algorithms to find such a permutation [1, 10]. In particular, Jordan et al. [10] observe simple permutation alone does not yield a low loss barrier which is caused by a phenomenon called variance collapse. They introduce the REPAIR method by rescaling the parameters in the interpolated networks [10]. This paper uses REPAIR to permute our trained models.

*Feature Visualization*. Visual interpretability methods provide an intuitive way to understand how neural networks function and make decisions. A saliency map highlights the area that has the most influence on the final model prediction [15]. Activation maximization finds input images that maximize the neuron activity by optimization [5, 14]. Through visualization, Li et al. [12] show that the representations learned by different networks converge to a set of features that are individually similar between networks [12]. In addition to visual-only qualitative analysis, Nanfack et al. [13] proposed Kendall-$\tau$ and CLIP-$\delta$ to characterize the similarity of images extracted from two models [13]. In this study, we visualize each layer of the trained model with top activated images from the training dataset and evaluate using Kendall-$\tau$ and CLIP-$\delta$ against the top images of each channel.

## 2   Experiments and Results

Our experiments focus on the AlexNet [11] and ResNet [8] architectures trained on the ImageNet dataset [2]. This choice is made largely because feature visualization methods work well and have been extensively studied for this model and dataset [11, 12].

We train five AlexNet models to use in our merging experiments. All model weights are randomly initialized, and each model is trained with 90 epochs. The validation accuracy for each model is $56.74\%, 56.25\%, 56.27\%, 56.45\%$, and $56.27\%$. We also trained two AlexNet models with stratified ImageNet dataset, one model trained with $80\%$ of 500 first image classes and $20\%$ of the last 500 image classes, and another model trained on the rest data. Three ResNet18 [8] models are trained as well. Visualization results of standard trained AlexNet are included in the main paper. More results for stratified AlexNet and ResNet are provided in the Appendix, which follows similar trends.

### 2.1   Visualizing Permutation Based Model Merging

We take two trained AlexNet models, with one (Model 2) permuted to align with the other (Model 1). Then we feed the training dataset to both models and extract the 10 images that activate each convolutional channel most (top-10 feature visualizations). For a selected channel, top images of

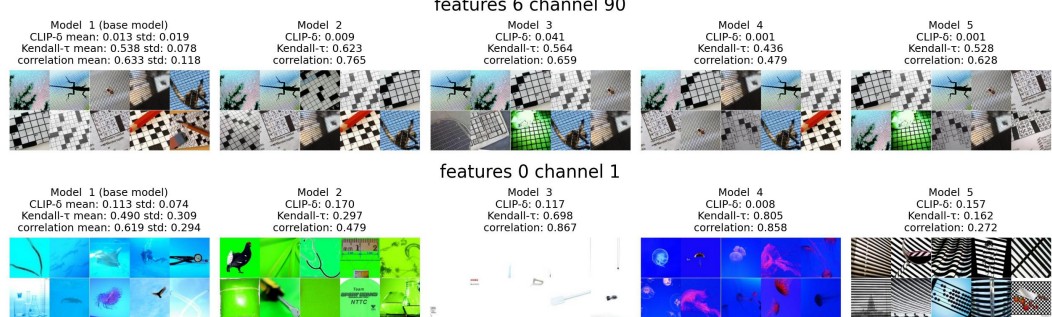

Figure 2: **Top Images Extracted from 5 AlexNet Models.** Models 2 to 5 are all permuted to align with model 1. The **top** row shows the case that all top images from the 5 models are visually similar, and the **bottom** row is the case where all top images are not similar. The metrics also indicate the same as all with high Kendall-$\tau$ and correlation values and low in CLIP-$\delta$. The bottom figure shows the case where images from all models are not similar.

original model 1, model 2 (before permutation), permuted model 2, permuted and merged model, and direct average merge (without permutation) are shown in Fig 1.

**Observation 1: Many but not all permutation-matched features are similar**. We informally define *feature visualization similarity (FV)* as a visual observer identifying the same structures and patterns (and sometimes the same images) between two visualizations of two different channels (typically found in two different models). We observe by looking through the visualizations over many channels that the permuted model 2 feature visualizations are often but not always similar to those from model 1 as illustrated in Fig 1 (bottom). Thus despite being trained from different initialization, the training will often find similar features. This also matches observations in [12].

We now turn to analyzing the feature visualization of the merged model.

**Observation 2: Merged model feature visualization is dependent on the source models and whether there is an FV similarity after permutation matching**. Analyzing the feature visualizations from many channels we make the following observations. In the case that the permuted model has similar top-activated images, the top images from the merged model are also similar. This is illustrated on the top in Fig 1 where the permuted model 2 shows similar features to model 1. However, when the images from 2 models are not similar after permutation (e.g. bottom of Fig 1), we observe different behavior on merged models top images. These behaviors were observed by inspection of all channels in AlexNet to fall into three general categories:

1. Images from one model (either model 1 or 2) dominate the top images of the merged model.
2. Images from both models appear in the top images of the merged model.
3. Most top images of merged models do not belong to any of the 2 models.

## 2.2 Aligning Multiple Models to One

Since the matching of features is so critical to the visualization observed in the merged model, we ask whether the mismatched cases after permutation occur randomly or if they represent rare features in model 1 that are hard to match in model 2. To study this we perform the matching process multiple times with one fixed model being matched.

Specifically, we permute the 4 of our AlexNet models (all trained from different initialization) to align with model 1. Top activated images are extracted for all 5 models.

**Observation 3: Certain features of the reference model are challenging to match in terms of FV similarity**. Analyzing all channels in all layers of AlexNet we have the following cases (see Fig 2):

1. For a channel all 5 models have very similar top images (occurring $(5 - 10\%)$)
2. For a channel most models have similar top images except one or two (occurring $(35 - 50\%)$).
3. For a channel top images from all 5 models are dissimilar $(40 - 60\%)$

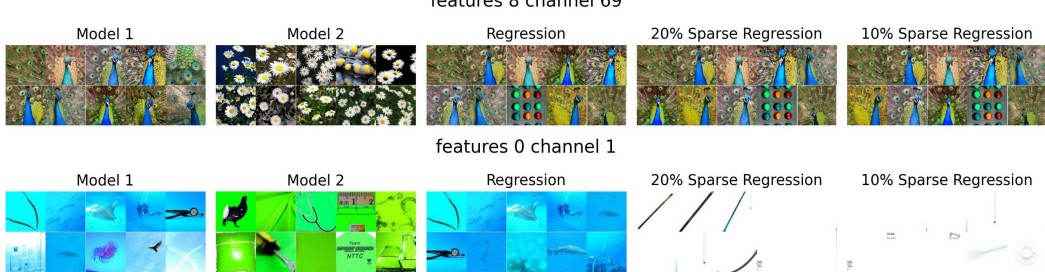

Figure 3: **Top images aligned after applying regression model.** In the **top** row, even with only 10% sparse regression weights, the images are still similar. However, there are also some cases where after applying sparse weight, the images changed. Examples are shown in **bottom** row.

We also study the mean and standard deviation of the metrics we use. It is clear that channels with high means (low for CLIP-$\delta$ scores) and low standard deviations usually have very similar top-activated images. On the other hand, if we have high standard deviation and low means, visually, the top images vary a lot.

The results indicate that certain features are easily learned by the models, while some features are hard to learn. This is what we hypothesize, and it leads to cases of dissimilar FV after permutation (Observation 1).

### 2.3 Aligning Models Using Linear Regression

We observe that although per neuron or channel, feature visualization is based on the hypothesis that each channel represents a particular feature of the data, a given useful feature may be represented by a single channel in one network but can be represented in a distributed way by another network.

We take all the features in a given layer of model 2 and regress them onto the features of model 1, building a separate regression model for each feature in model 1. Specifically, we get the activation of each convolutional channel from both models with 5000 images randomly selected from the ImageNet dataset, i.e., 5 images per class. Then for each layer, we find a linear regression model, such that for activation values of each channel in model 1, it is approximated by a linear combination of activation values from all channels in model 2. Afterward, we feed the entire dataset to model 2, apply the regression model on the activations, and extract the top 10 activated images of each channel.

**Observation 4: For each feature in the model there is a (often sparse) linear combination of features in model 2 that gives an FV similarity**. In Fig 3, we see the results of visualizing the regression model outputs at each layer. The results show that top-activated images are now visually aligned. Furthermore, we study the sparsity of the weights we got from the regression model. As shown in Fig 3, we can see the top images do not change a lot when we only select the weights with top magnitude and set the rest to 0. This shows that the weights that impact the alignment are sparse. However we do find that for certain channels (particularly in the early layers), the top activated images change when we use less weights, and these channels are also visually hard to align channels when we compare the 5 models (Fig 3).

Our results suggest that permutation-based merging can be substantially limited since it is too restrictive to assume a one-to-one matching between features. Indeed recent model merging methods specifically aim to address this restriction [16, 9], showing promising new results. Our work helps to provide additional motivation for these new methods.

## 3  Conclusion

By permuting trained models and visualizing top-activated images, we have gained valuable insights into the internal behavior of these models. Our findings visually demonstrate the effectiveness of model alignment by permutation and provide some insights into linear mode connectivity. Additionally, the use of linear regression to align models has shown promising results, and the sparsity of effective weights suggests an inner connection of the channels.

## Acknowledgments and Disclosure of Funding

This research was partially funded by NSERC Discovery Grant RGPIN- 2021-04104 and FRQNT New Researcher Grant. We acknowledge resources provided by Compute Canada and Calcul Quebec.

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

# Appendix

## A    Additional Results for Standard Trained AlexNet Models

We trained 5 AlexNet models on full ImageNet Dataset for 90 epochs with SGD optimization. Learning rate starts at 0.01 and decays by a factor of 10 every 30 epochs. All experiments were conducted using a single V100 GPU with 16GB of memory.

### A.1    Permutation and Merge Results

We take 2 trained models and apply REPAIR algorithm to permute one model based on the other for alignment. Next, we merge the models using multiple interpolation coefficients $\alpha$ ranging from 0 to 1, with a step size of 0.05 and evaluate the merged models performance on the validation dataset. Additionally, we implement the algorithm to rescale the weights in merged model (referred to as **Repaired model** in the Figures) and record the corresponding validation accuracy. Models are also directed merged without permutation as base line for comparison. Fig 4 shows merge results for standard trained AlexNet model. Models are trained 90 epochs with SGD optimizations. Validation accuracies improve greatly compared to direct merging. However repaired merge models do not perform significantly better.

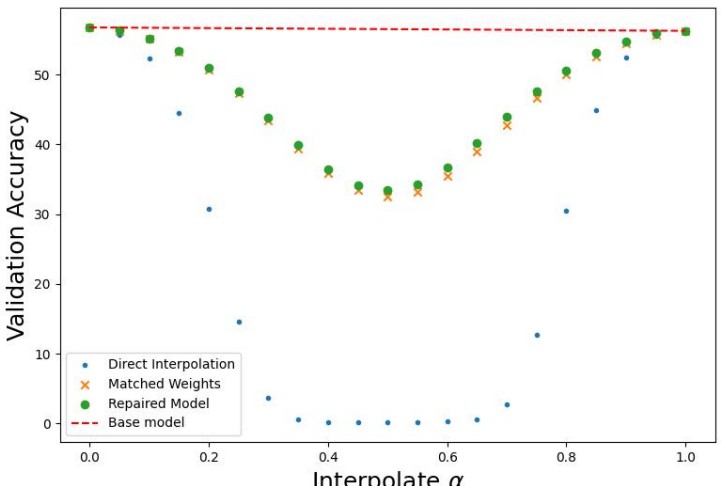

Figure 4: Merge Results for Standard Trained AlexNet.

## A.2 Top Images for Permutation Based Model Merging

Model 2 is permuted to align with model 1. Both direct merge and permuted merged models are studied, as well as repaired merge model. Top activated images from original model 2 before permutation is also included. Visually for most channels, top images are getting more similar after permutation, and the merging model shares the same similarity with slightly change in repaired model. (Fig 5) In the case where top images from permuted model are not similar to those from model 1, we get different cases for merged model images. See Fig 6, 7, 8.

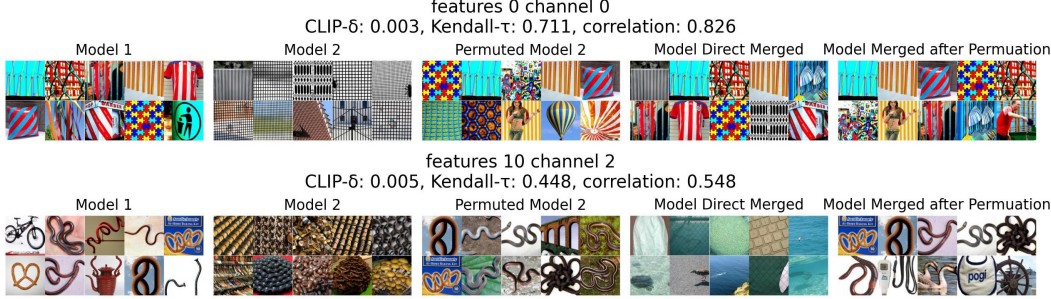

Figure 5: Similar top images from model 1 and permuted model 2 with similar results from merged models.

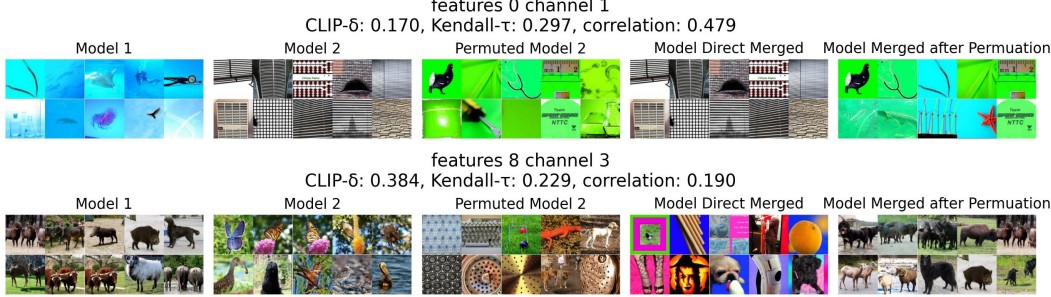

Figure 6: Dissimilar top images from model 1 and permuted model 2 with images from one model dominate the top images of merged mode.

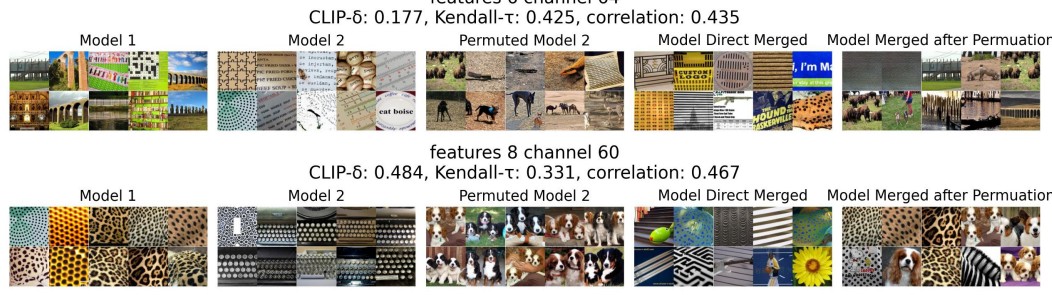

Figure 7: Dissimilar top images from model 1 and permuted model 2 with images from both models appear in the top images of merged model.

## A.3 Top Images for Aligning Multiple Models to One

We permuted 4 models against model 1. Visually, we can see that the similarity of the top images from 5 models varies from channel to channel. Some channels have similar images from all 5 models, and some have images totally dissimilar from all 5 models. There are also channels with similar images from 3 model and another type of similar images from 2 other models. Furthermore, We compute

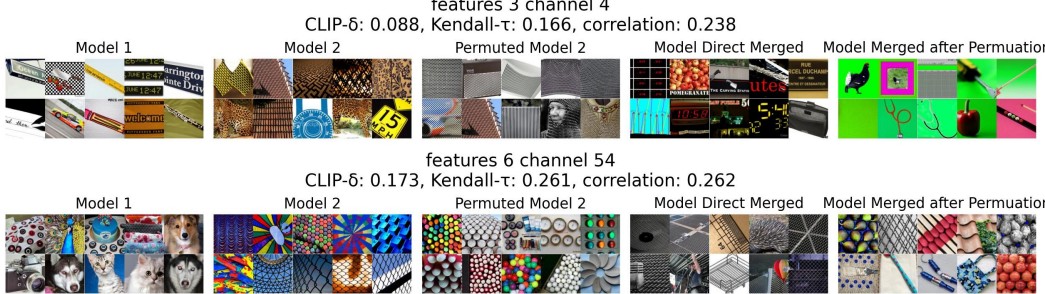

Figure 8: Dissimilar top images from model 1 and permuted model 2 with most top images of merged model do not belong to any of the 2 models

the CLIP-$\delta$, Kendall-$\tau$ and correlation metrics for each channel compared to model 1. Means and standard deviations are also shown. These metrics match our visual intuition. For individual model, a higher Kendall-$\tau$ and correlation value or a lower CLIP-$\delta$ usually means top images are more similar to those in model 1. Cross all 5 models, a higher mean with lower standard deviation implies similar images, and a lower mean with higher standard deviation show dissimilarity. Feature visualization results from selected channels are shown in Fig 9.

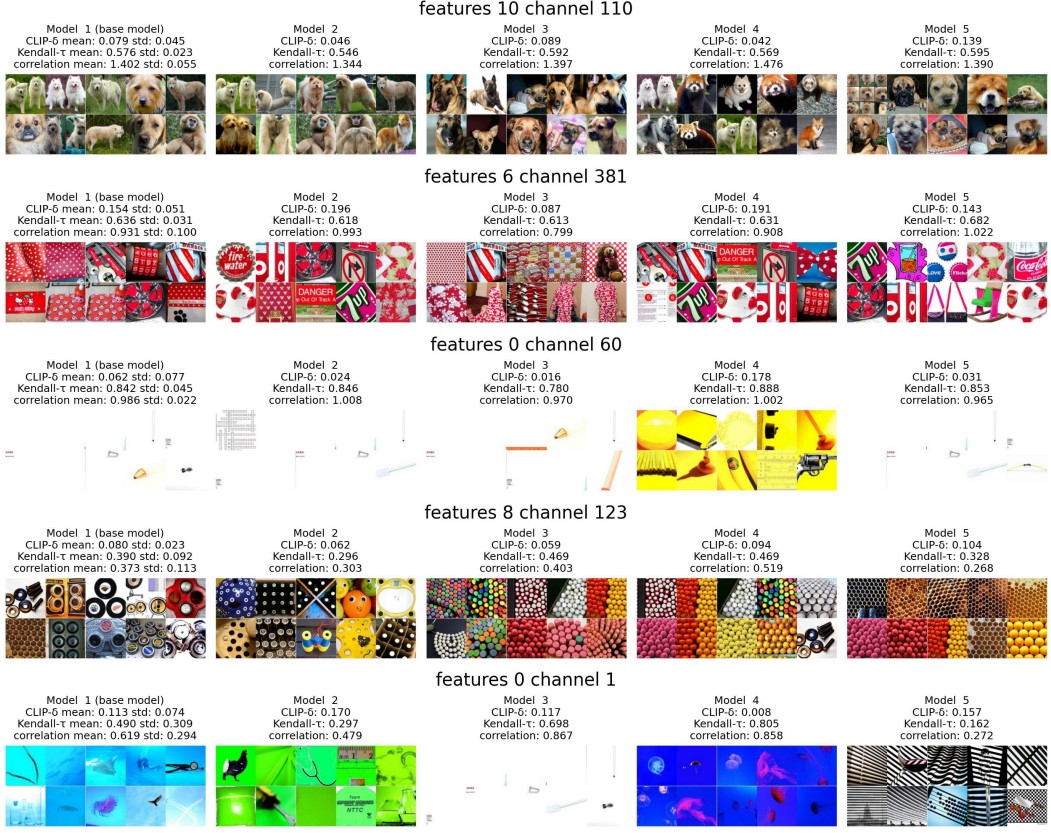

Figure 9: Top activated images from 5 AlexNet models. Model 2 to 4 are permuted against Model 1. Different cases described in the main paper is shown here. Similarities of top activated images across 5 models varies.

## A.4 Top Images for Linear Regression

We create a linear regression model, which take all the features in a given layer of model 2 and regress them onto the features of model 1. We compute the activation of each convolutional channels from both models with 5000 images randomly selected from ImageNet dataset, i.e., 5 images per classes. Then for each layer, we implement a linear regression model, such that for activation values of each channel in model 1, we have a linear combination of all channels in model 2 to approximate them. Afterwards, we feed the entire dataset to model 2, apply the regression model on the activations, and extract the top 10 activated images of each channels. Results for linear regression experiments are in Fig 10. For majority of the channels, top activated images are aligned with even top 5 weights used.

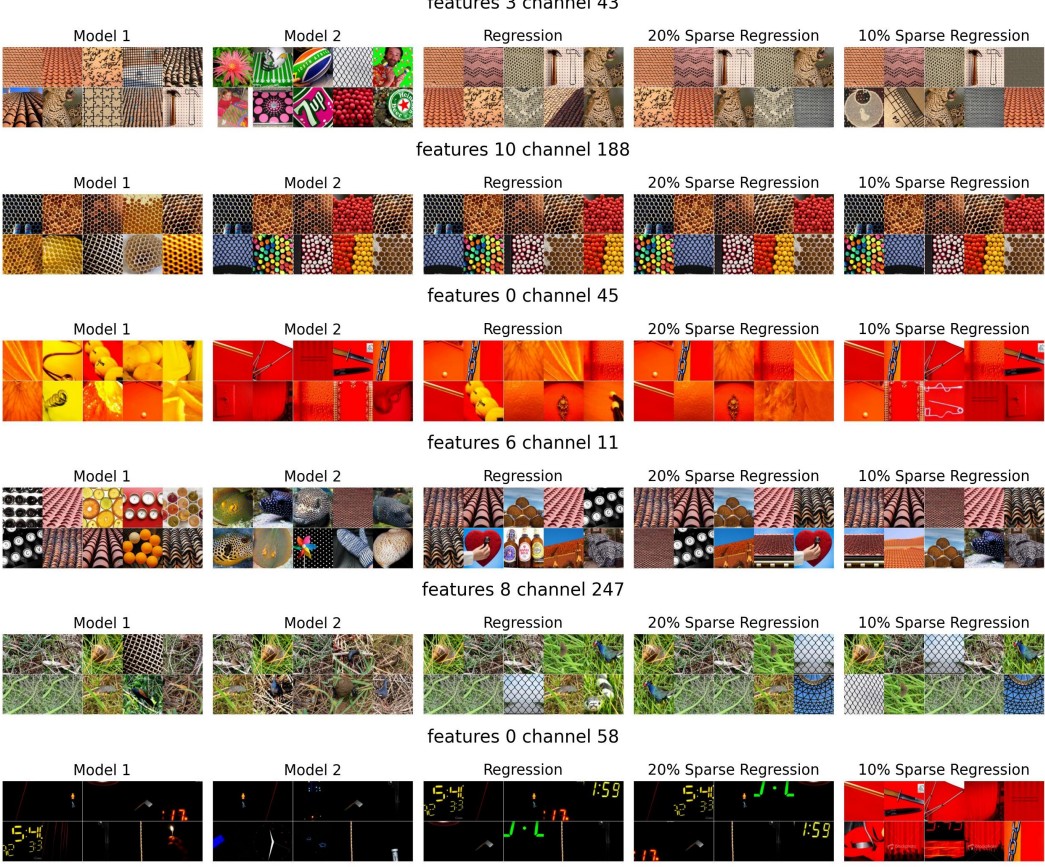

Figure 10: Top activated images after apply linear regression. In most channels features are similar with only 10% sparse regression weights.

# B   Additional Results for AlexNet Models Trained with Stratified Dataset

We trained 2 AlexNet model with stratified dataset. One model is trained with $80\%$ of 500 first image classes and $20\%$ of the last 500 image classes, and another model is trained on the rest data. Both models are trained with SGD optimizer for 90 epochs. Learning rate starts at 0.01 and decays by a factor of 10 every 30 epochs.

## B.1   Permutation and Merge Results

Fig 11 shows merge results for AlexNet model trained with unbalanced data. We carry the same experiment as described in **A.1**. Validation accuracies improve greatly compared to direct merging. Repaired merge models also perform better than the permuted only models.

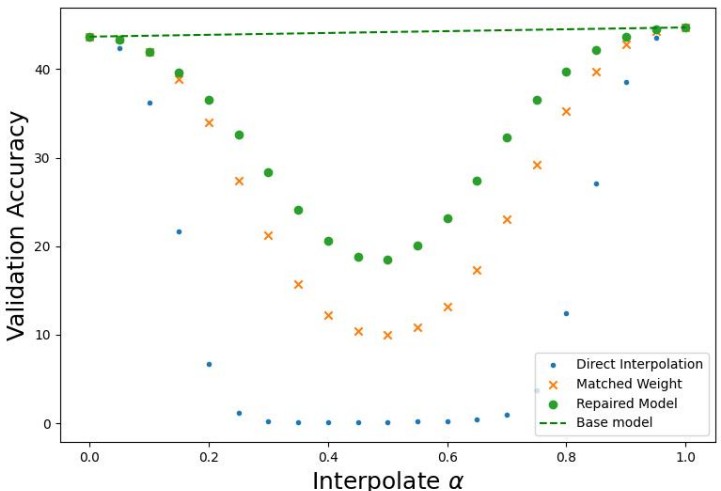

Figure 11: Merge Results for AlexNet Trained with Stratified Dataset.

## B.2 Top Images for Permutation Based Model Merging

Model 2 is permuted to align with model 1. Both direct merge and permuted merged models are studied, as well as repaired merge model. Top activated images from original model 2 before permutation is also included. Fig 12 shows some selected results. We observe similar pattern as standard trained models. Although due to the unbalanced dataset, the visualization extracted shows less similarity compared to the models trained with full dataset.

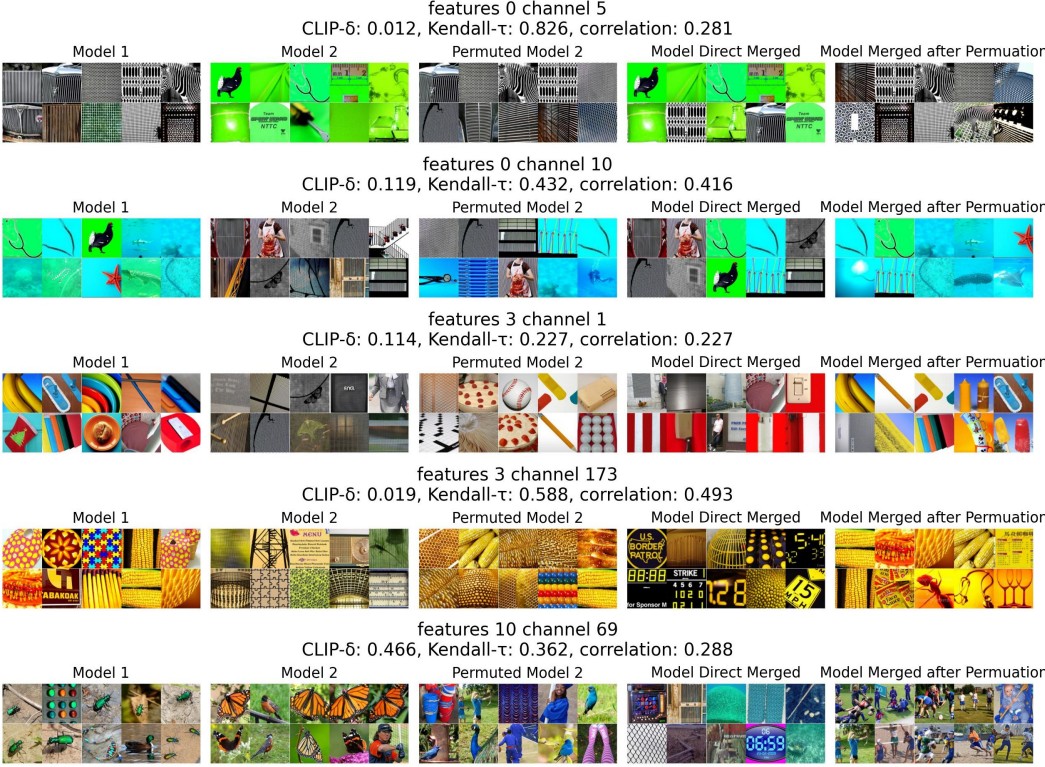

Figure 12: Top activated images from selected channels for AlexNet models trained with stratified dataset. Results are similar to models trained with standard dataset. It is also observed that features are harder to align compared to standard trained models. This is probably because of the unbalanced training dataset.

# C   Additional Results for ResNet18 Models

We trained 3 ResNet18 models. Each models are trained on full ImageNet Dataset for 100 epochs with SGD optimization. Learning rate starts at 0.1 and decays by a factor of 10 every 30 epochs.

## C.1   Permutation and Merge Results

Fig 13 shows merge results for ResNet model. Same experiments as in **A.1** are performed. We observe some accuracy improvement with permuted only models, especially when $\alpha$ is less than 0.4 and greater then 0.6. After rescaling, the validation accuracies are improved significantly.

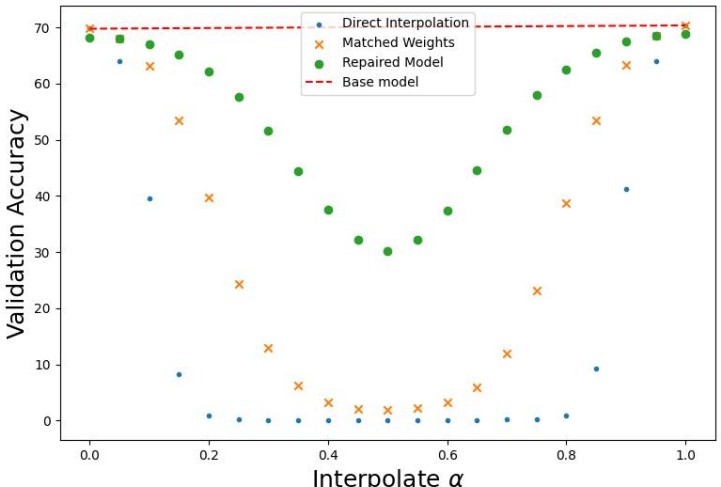

Figure 13: Merge Results for Standard Trained ResNet18.

Due to the ResNet18 model architecture and permutation algorithm, the feature visualization experiments are not performed with each convolutional layer, but with each residual block instead. There total 8 residual blocks in a ResNet18 model. Similar to previous experiments, we visualize top-10 images that activate each block most in following models: model 1, model 2, permuted model 2, directly merged model and permuted and merged model. Results are shown in Fig 14. We also observe similar pattern as AlexNet models are observed.

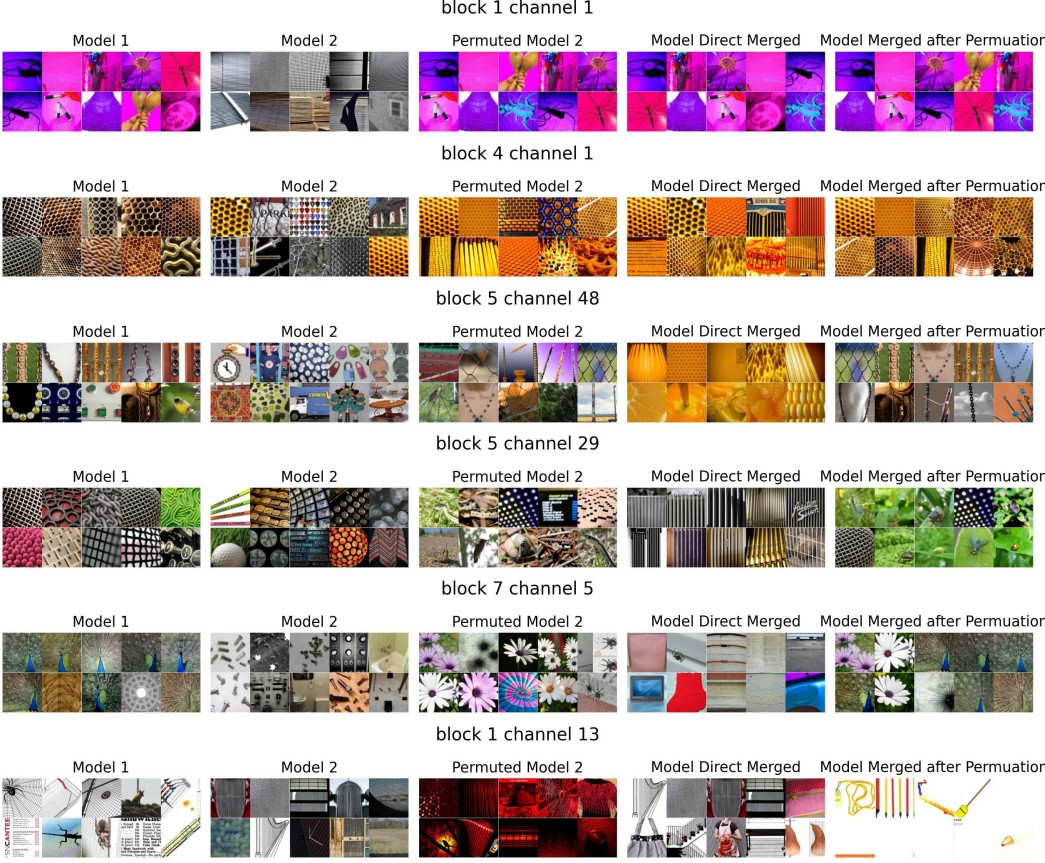

Figure 14: Top images for 2 ResNet18 models. Model 2 is permuted to align with model 1. Top images from original model 2 and direct merged model are also shown. Top images shows similarities between model 1 and permuted model 2. In the case of dissimilarity, we observe that top images from merged model after permutation can have images from both model 1 and permuted model 2 or from none of them, which shares the same pattern as AlexNet models.

### C.3 Top Images for Aligning Multiple Models to One

We then permute one more model (model 3) and visualize top-10 images that activate each block most in each model. Results are shown in Fig 15. We can see similar pattern as AlexNet models are observed.

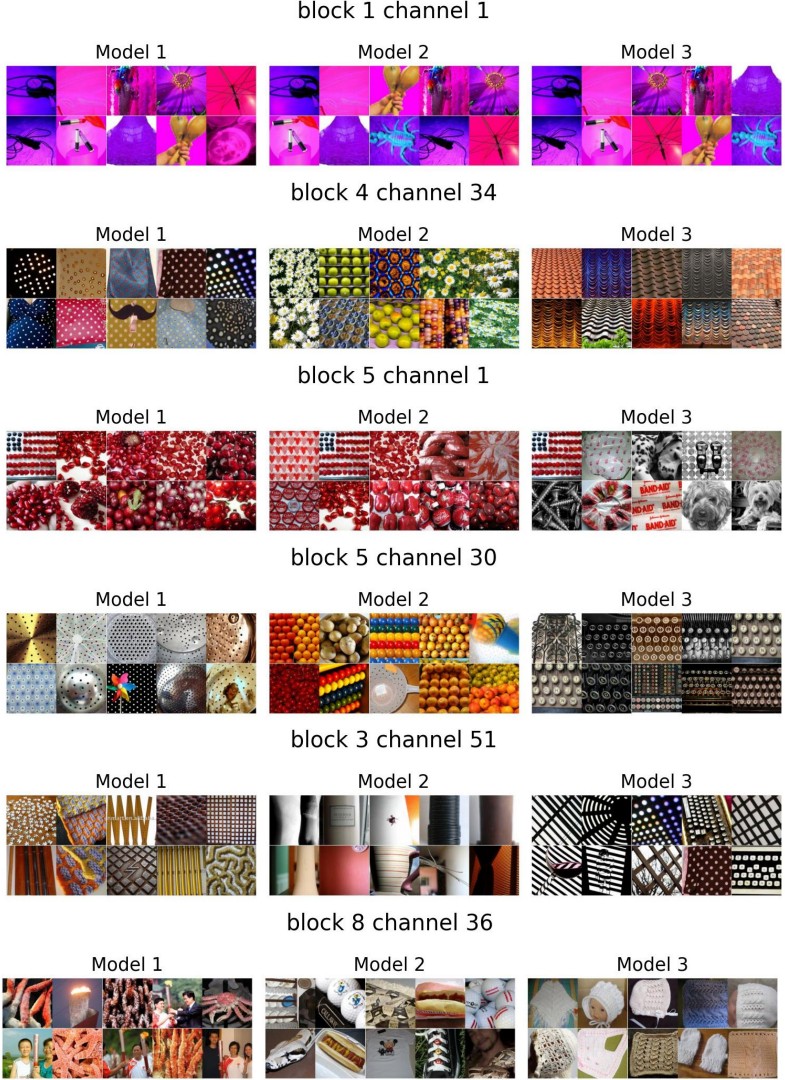

Figure 15: Top activated images from 3 ResNet18 models. Model 2 and 3 are are more aligned to model 1 after permutation. In some cases, we have only one model (2 or 3) visually aligned to model 1. There are also cases that neither of the 2 models shows FV similarity after permutation.

## D Limitations and Future Work

This study is mainly empirical, and the metrics we use do not clearly differentiate each case. Future research could include quantizing the similarity of the aligned channels and potentially applied in model merging. Furthermore, the sparse linear combination of the permuted channels representation may also be a promising approach of merging two models.

