# OpenReview forum: "Understanding Permutation Based Model Merging with Feature Visualizations"
_NeurIPS.cc/2024/Workshop/UniReps — UniReps_

### Official Review · Reviewer_5Dhs · 2024-10-01
**Promising path with opportunities for refinement**

**Rating:** 4
**Confidence:** 3

**Review:**

## Summary

The paper uses feature visualisation methods to explore how different randomly initialised image classifiers, trained on the same dataset, activate similar features and how these activations change when models are aligned using permutation or linear regression. Moreover, the paper produces visualisations that illustrate how merging of original and aligned models influences the features each channel responds to. The analysis reveals that the different models often produce channels that detect similar features, but this alignment is not guaranteed.

## Strengths and Weaknesses

Strengths:
- Feature Visualisations are a promising tool to understand when model alignment work and when it does not.
- The visualisations with increasing sparser regressions can provide a good intuition on how much sparsity is needed and what are the effects of having more or less sparsity.

Weaknesses:

- Both the abstract and the conclusion are very vague in terms of what can be concluded from the results.
- Only feature visualisations are shown. These are good to highlight particular interesting cases, but they fail to communicate the statistic relevance of the visualisations and what is the overall model behaviour. Including aggregate metrics like CLIP-$\delta$ and Kendall-$\tau$ across multiple layers would strengthen the statistical grounding of the results.
- Figure 3 should also show the model 2 after permutation so that the reader can compare that method with the regression.

## Questions

- The figures caption and legend could be more clear. What does the number of “features” means in the title? In Figure 1, the CLIP-$\delta$ and the Kendal-$\tau$ are a measure between which images?


Soundness: 2

Presentation: 2

Contribution: 2

---

### Official Review · Reviewer_1cHt · 2024-10-03
**This article is mainly based on experiments and is meaningful.**

**Rating:** 6
**Confidence:** 3

**Review:**

The work focuses on visualizing model behaviors using feature visualization techniques on CNNs. The experimental results give insights into how models align after permutation and what impact this has on their performance and feature extraction.
1. The architecture of AlexNet and ResNet is investigated. It is better to include transformer-based architectures.
2. While the paper effectively demonstrates feature similarity after model permutation, the broader implications of this work for practical applications such as transfer learning or federated learning are absent.

---

### Official Review · Reviewer_msHQ · 2024-10-06
**Interesting research questions, but current methodology/results are unconvincing.**

**Rating:** 6
**Confidence:** 4

**Review:**

I believe the research direction and approach is novel, and the paper is above the bar for a workshop submission in writing quality. The results, even for a workshop submission, were not very convincing to me however and I would suggest to the authors that their results would benefit much more from a more rigorous and quantitative analysis moving forward.

Strengths:
- Model merging itself is quite opaque still, and some of the author's questions are indeed of interest.
- We can can expect learned features/filters to differ between different models trained with different random initialization (even after weight or activation matching/permutation, which is not perfect) to differ, and of course expect merged models to learn different features, and I believe the best experimental directions proposed in this work align with the question of how these differ - I myself would love to see convincing results/analysis that would explain these areas further.
- Decent summary (for workshop paper) of LMC/permutation background, although I believe the wording of their main research question on line 26 should be improved.

Weaknesses
- The features/filters of a model before/after applying a permutation *should not differ* by definition, and likewise the images with maximum activations should also not differ, however we can expect the specific channel ordering of the filters learned to change. The author's results images focus on single channels unfortunately (e.g. Figure 1), and might even mislead some to believe model 2 v.s. model 2 permuted learn different filters. Only showing one channel gives a very incomplete intuition as to how similar the features learned by one model are v.s. another. This is true even after permutation, as weight matching/activation matching only approximate a good permutation (not perfect).
- Analysis is effectively qualitative (see below on metrics). The authors judge if models learn similar features/filters solely based on the top-10 images that maximally activate neurons (filters in CNN) in the corresponding models. The authors should take care to note however that even if two neurons share the same top-10 images, it is not necessarily the case that they have learned the same representation, unlike they suggest. This is however evidence that would support such a hypothesis, and I would suggest to the authors that it would be much more convincing if paired with other more quantitative evidence such as existing methods of comparing neural representations (e.g. "Similarity of Neural Network Representations Revisited", Kornblith et al., ICML 2019).
- The only metrics the authors use are used to compare images (i.e. the images that have maximum activation), rather than filters/features learned in the models, which does not address the issue I have pointed out to above, but is much better than only including visuals of channels of images. It is not clear to me however if the metrics are calculated over all channels or only a single channel.
- The images themselves are far too small to be the main results of the paper, it's very hard to make out individual images.
- Kind of strange to be using AlexNet for anything in 2024, although given the ResNet results I don't think this is a big weakness. I would suggest the authors find further, more modern architectures to compare however, although I realize weight/activation matching work better with wider models.
- I looked at some of the appendix out of curiosity, and didn't understand why there were class-unbalanced training results there - would suggest motivating those results even if they are in the appendix.

---

### Official Review · Reviewer_peWj · 2024-10-07
**Understanding LMC and permutation matching by visualizing activation maps**

**Rating:** 6
**Confidence:** 4

**Review:**

The study of the permutation matching of two models by inspecting the feature activation map.

### Strengths

1. The paper is mainly easy to follow, and the idea of visually comparing the activation maps is intuitive and interesting.

2. The authors make interesting observations using the feature maps, which I think is worth discussing in the workshop. However, there are still shortcomings which authors can improve on for the next version of the manuscript.

### Weaknesses

1.
> We also trained two AlexNet models with stratified
ImageNet dataset, in which one model trained with 80% of 500 first image classes and 20% of the last
500 image classes, and another model trained on the rest of the data.

The intuition behind this split is not clear.

2. AlexNet used in the experiments is a small model, and observations may not hold true for larger models.

3. Figure 1 is not clear, columns should be labeled. Permuting model 2 (column 2) should not change the top-10 activated images, so column 2 and column 3 (model 2 permuted) should have same images (but different order). However, that is not case and it is not clear why.

---

### Decision · Program_Chairs · 2024-10-10

**Decision:**

Accept

**Comment:**

In light of the reviewers' feedback and relevancy of the submission, we are pleased to accept this paper for presentation at UniReps 2024. We kindly ask the authors to incorporate the reviewers' suggestions and feedback in the final camera-ready version of the manuscript.